# The Preventive Effect of Urinary Trypsin Inhibitor on Postoperative Cognitive Dysfunction, on the Aspect of Behavior, Evaluated by Y-Maze Test, via Modulation of Microglial Activity

**DOI:** 10.3390/ijms25052708

**Published:** 2024-02-26

**Authors:** Eun-Hwa Cho, Chi-Bum In, Gyu-Won Lee, Seung-Wan Hong, Eun-Hye Seo, Won Hyung Lee, Seong-Hyop Kim

**Affiliations:** 1Department of Infection and Immunology, School of Medicine, Konkuk University, Seoul 05030, Republic of Korea; eunhwa099@naver.com; 2Department of Anesthesiology and Pain Medicine, Konyang University Hospital, Daejeon 35365, Republic of Korea; cb523@naver.com (C.-B.I.); dlrbdnjs11@naver.com (G.-W.L.); 3Myunggok Medical Research Institute, College of Medicine, Konyang University, Daejeon 35365, Republic of Korea; 4Department of Anesthesiology and Pain Medicine, Konkuk University Medical Center, School of Medicine, Konkuk University, Seoul 05030, Republic of Korea; 20150077@kuh.ac.kr; 5Korea mRNA Vaccine Initiative, Gachon University, Incheon 21936, Republic of Korea; gmreo@naver.com; 6Department of Anesthesiology and Pain Medicine, College of Medicine, Chungnam National University, Daejeon 35015, Republic of Korea; whlee@cnu.ac.kr; 7Department of Anesthesiology and Pain Medicine, Chungnam National University Hospital, Daejeon 35015, Republic of Korea; 8Department of Medicine, Institute of Biomedical Science and Technology, School of Medicine, Konkuk University, Seoul 05030, Republic of Korea; 9Department of Medical Education, School of Medicine, Konkuk University, Seoul 05030, Republic of Korea

**Keywords:** ulinastatin, urinary trypsin inhibitor, postoperative cognitive dysfunction, microglia

## Abstract

This experimental study was designed to evaluate the effect of ulinastatin, a urinary trypsin inhibitor, on postoperative cognitive dysfunction (POCD) in rats under general anesthesia with isoflurane, on the aspect of behavior, as evaluated using a Y-maze test and focusing on microglial activity. Ulinastatin (50,000 U/mL) and normal saline (1 mL) were randomly (1:1) administered intraperitoneally to the ulinastatin and control groups, respectively, before general anesthesia. Anesthesia with isoflurane 1.5 volume% was maintained for 2 h. The Y-maze test was used to evaluate cognitive function. Neuronal damage using caspase-1 expression, the degree of inflammation through cytokine detection, and microglial activation with differentiation of the phenotypic expression were evaluated. Twelve rats were enrolled in the study and evenly allocated into the two groups, with no dropouts from the study. The Y-maze test showed similar results in the two groups before general anesthesia (63 ± 12% in the control group vs. 64 ± 12% in the ulinastatin group, *p* = 0.81). However, a significant difference was observed between the two groups after general anesthesia (17 ± 24% in the control group vs. 60 ± 12% in the ulinastatin group, *p* = 0.006). The ulinastatin group showed significantly lower expression of caspase-1. Pro-inflammatory cytokine levels were significantly lower in the ulinastatin group than in the control group. The ulinastatin group had a significantly lower microglial activation (41.74 ± 10.56% in the control group vs. 4.77 ± 0.56% in the ulinastatin, *p* < 0.001), with a significantly lower activation of M1 phenotypes (52.19 ± 7.83% in the control group vs. 5.58 ± 0.76% in the ulinastatin group, *p* < 0.001). Administering ulinastatin before general anesthesia prevented neuronal damage and cognitive decline after general anesthesia, in terms of the aspect of behavior, as evaluated by the Y-maze test. The protective effect of ulinastatin was associated with the inhibition of microglial activation, especially the M1 phenotype.

## 1. Introduction

The need for anesthesia and surgery in elderly patients has increased with the increase in life expectancy. Although the technical evolution of anesthesia and surgery, and the physical status of the elderly patients, are much improved, postoperative complications frequently occur in the elderly patients. Postoperative cognitive dysfunction (POCD) is a postoperative complication that frequently occurs in elderly patients [1]. POCD is defined as a new-onset cognitive impairment after anesthesia and surgery [2]. It may persist for weeks or month postoperatively. It seriously affects postoperative recovery and quality of life. To diagnose POCD, pre- and postoperative tests for cognitive performance are required. Other disorders with postoperative cognitive change, such as definite anatomical disorder of the central nervous system, metabolic disorder, and electrolyte imbalance, should be differentially diagnosed, using proper diagnostic tools [3]. Bedsides advanced age, patients with cerebral, cardiac or vascular disease, preoperative cognitive impairment, low education, history of alcohol abuse, reoperation, longer duration of anesthesia, perioperative hypoxia, etc., have been known to be the risk factors of POCD [4]. However, the mechanisms underlying POCD have not been completely elucidated, although various mechanisms for POCD, such as neuroinflammation, oxidative stress, autophagy disorder, and impaired synaptic function, have been proposed. Nevertheless, accumulating evidence indicates that neuroinflammation is one of most important mechanisms of POCD [1]. At the occurrence of neuroinflammation, microglia are resident macrophages in the brain and the spinal cord, and are the first responders against inflammation in the central nervous system (CNS) [5].

Ulinastatin, a urinary trypsin inhibitor, is purified from fresh healthy urine [6]. It inhibits protease activity and has anti-inflammatory characteristics [7]. Therefore, it has been widely used for the treatment of inflammatory diseases, such as acute pancreatitis, and sepsis, although it is available only in China, India, Japan, and Korea [6,8,9]. Considering the anti-inflammatory properties of ulinastatin, it might prevent POCD by modulating neuroinflammation via microglial activity in the CNS. However, no related studies have been conducted.

We hypothesized that the preoperative administration of ulinastatin, which inhibits inflammatory response, may be beneficial in preventing POCD, by modulating neuroinflammation via microglial activity. The study was designed to evaluate the effect of ulinastatin on POCD in rats under general anesthesia with isoflurane, focusing on microglial activity.

## 2. Results

Twelve rats were enrolled in the study and evenly allocated into the two groups, with no dropouts from the study.

The alternation ratios of the two groups, using the Y-maze test, were similar before general anesthesia (63 ± 12% in the control group vs. 64 ± 12% in the ulinastatin group, *p* = 0.81); however, a significant difference was observed after general anesthesia (17 ± 24% in the control group vs. 60 ± 12% in the ulinastatin group, *p* = 0.006). General anesthesia significantly reduced the alternation ratio in the control group (*p* = 0.005). However, the alternation ratios before and after general anesthesia in the ulinastatin group were similar (*p* = 0.58) (Figure 1).

The neuronal damage, expressed by caspase-1, was significantly different between the groups. The ulinastatin group showed a significantly lower expression of caspase-1 (12.91 ± 3.89% in the control group vs. 2.73 ± 0.28% in the ulinastatin group, *p* < 0.001) (Figure 2A). The ulinastatin group also showed a significant lower number of cells expressing caspase-1 (52.00 ± 10.86 in the control group vs. 4.60 ± 2.70 in the ulinastatin group, *p* < 0.001) (Figure 2B).

The pro-inflammatory cytokines, tumor necrosis factor-α (TNF-α) and interleukin-1β (IL-1β), had significantly lower values in the ulinastatin group than in the control group, and the anti-inflammatory cytokine, IL-10, had significantly higher values in the ulinastatin group than in the control group (Table 1). However, IL-4 levels were not significantly different between the two groups (Table 1).

The ulinastatin group had a significantly lower activation of microglia (41.74 ± 10.56% in the control group vs. 4.77 ± 0.56% in the ulinastatin group, *p* < 0.001) (Figure 3). The ulinastatin group also had a significantly lower activation of M1 phenotypes (52.19 ± 7.83% in the control group vs. 5.58 ± 0.76% in the ulinastatin group, *p* < 0.001) (Figure 4). However, the activation of M2 phenotypes in both groups was similar (8.63 ± 7.29% in the control group vs. 11.66 ± 1.49 in the ulinastatin group, *p* = 0.39) (Figure 4).

## 3. Discussion

The present study demonstrated that ulinastatin administration before general anesthesia prevented cognitive decline after general anesthesia, in terms of the aspect of behavior and as evaluated by the Y-maze test. Administering ulinastatin before general anesthesia reduced neuronal damage, as evaluated using caspase-1. The effect of ulinastatin was associated with the significantly lower activation of microglia, mainly the M1 phenotype, and significantly lower expression of pro-inflammatory cytokines.

In the present study, cognitive function after general anesthesia in the control group decreased from 63 ± 12% to 17 ± 24%. This indicates that general anesthesia severely impacted cognitive function. Considering that rats aged 18 months old were used in the present study and that the age of rats represents humans aged 45 years old, the impact of general anesthesia on cognitive function might be more severe as they age. This finding is supported by previous studies [10]. Cognitive dysfunction after general anesthesia results in increased morbidity and mortality, especially in elderly patients [1,11]. Therefore, prevention of POCD is important to achieve better recovery after surgery and improved quality of life after discharge from the hospital.

Although the mechanisms underlying POCD have not been elucidated, numerous studies have demonstrated that POCD is associated with neuroinflammation [12,13]. The blood–brain barrier (BBB) plays a critical role in brain homeostasis. The BBB is composed of three cell types: endothelial cells, astrocytes, and microglia. The BBB supplies nutrients to the brain and shields the brain from any harmful components in the blood [14]. An insult of any cause to the brain induces neuroinflammation, resulting in dysfunction of the BBB [15]. BBB dysfunction leads to cognitive dysfunction [2,16]. Microglia, one of the cell types in the BBB, are the primary innate immune cells in the brain. This means that microglia are the first responders to neuroinflammation. However, there have been no reports on the effects of ulinastatin on microglia in POCD. Therefore, we focused on microglia in this study.

In this study, we evaluated caspase-1 expression to detect neuronal damage. Caspase-1 is an initiator of inflammation or programmed cell death in the CNS [17,18]. Increased caspase-1 expression is associated with various neurodegenerative diseases and BBB dysfunction [19]. Therefore, caspase-1 is a possible target for neurodegenerative diseases or BBB restoration [17]. Moreover, isoflurane anesthesia leads to caspase-1 activation [20]. The decreased expression of caspase-1 after the administration of ulinastatin in the present study indicates that ulinastatin might be a candidate of preventive or therapeutic agents for POCD.

The present study showed that the neuronal damage after general anesthesia was associated with a significant increase in the activation of total microglia. Activated microglia are known to differentiate into two different phenotypes, the M1 phenotype (a pro-inflammatory producer) and the M2 phenotype (an anti-inflammatory producer), depending on the environmental circumstances [21]. The M1 phenotype induces neuronal inflammation and toxicity. In contrast to the M1 phenotype, the M2 phenotype has anti-inflammatory and protective effects in neurons. In the present study, the administration of ulinastatin before general anesthesia significantly reduced the activation of the M1 phenotype, but not the M2 phenotype. This indicated that the protective effect of ulinastatin was associated with the inhibition of the M1 phenotype. This was supported by the patterns of cytokine release and inhibition of TNF-α and IL-1β release in the present study. Moreover, increased pro-inflammatory cytokine levels directly cause BBB dysfunction with BBB permeability and neuronal damage [22,23], although BBB function was not evaluated in the present study. Therefore, the inhibition of TNF-α and IL-1β release after ulinastatin administration in the present study might be evidence that ulinastatin protects BBB function in POCD.

Ulinastatin has demonstrated its anti-inflammatory effects in various diseases [24]. It has also demonstrated the preventive effects against CNS-related disorders in both laboratory and clinical settings [25]. Moreover, several recent meta-analyses have shown that ulinastatin is beneficial in preventing POCD, with inhibition of inflammatory cytokines, such as TNF-α, IL-6, C-reactive protein (CRP), and S100β, and augmentation of the anti-inflammatory cytokine, IL-10 [24,26,27,28]. However, the meta-analyses should be interpreted with caution because the studies were performed using different dosing strategies for ulinastatin in a limited population and area.

In the present study, the alteration ratio from the Y-maze test was used to evaluate cognitive function. Assessment of the alteration ratio is used to evaluate spatial memory and learning. This meant that the Y-maze test in the present study was useful for evaluating cognitive function, although it just assessed spatial memory and learning. Moreover, the Y-maze test does not necessitate rule learning, extensive handling, or repeated manipulations, compared with other tests for cognitive function [29]. If several other tests for cognitive function were simultaneously performed in the present study, more definite findings might be obtained. Actually, the Morris water maze test has been widely performed to evaluate cognitive function, instead of the Y-maze test. However, the Morris water maze test is also a tool used to assess behavioral tasks, like the Y-maze test. Moreover, the Morris water maze test is performed literally in the water. Unfortunately, the environment, in the water, is stressful to the experiment participants. A test conducted under stress might lead to wrong results [30]. Besides the Y-maze test, performance of another test for cognitive function, such as the Morris water maze test, might be another stress source. Therefore, we did not perform various tests for cognitive function in the present study.

The present study showed that the administration of ulinastatin before general anesthesia had a limited effect on the M2 phenotype because augmentation of the expression of the M2 phenotype with IL-4 was not statistically significant, although augmentation with IL-10 was statistically significant. The main target of ulinastatin appears to be the M1 phenotype, not the M2 phenotype, although increased IL-10 levels down-regulated the expression of the M1 phenotype [31,32]. However, several studies have shown that ulinastatin up-regulates anti-inflammatory cytokines [33], as shown in the aforementioned meta-analyses [32], with a significant increase in IL-10 release after ulinastatin administration. Therefore, further evaluation may be helpful to investigate the effect of ulinastatin on M2 differentiation.

This study has several considerations. First, the animals in the present study were anesthetized without performing any surgical procedure, and they exhibited neuronal damage and cognitive decline. This could misrepresent the idea that general anesthesia has a detrimental effect on cognitive function, although the effect of general anesthesia on cognitive function is controversial and complex. Several studies have shown that short- and long-term effects of general anesthesia on cognitive function differ [34,35]. Some studies have shown positive effects of general anesthesia on cognitive function [36]. In the present study, we did not perform hemodynamic, respiratory, or other monitoring. Although the two groups in the present study experienced a similar impact due to hemodynamic, respiratory, and other factors, except the general anesthesia itself, might play roles in the occurrence of POCD. Moreover, the evaluation of the pure effect of general anesthesia itself on POCD was impossible because of the effects of the surgical procedure, such as stress and inflammation, if such a procedure was performed in the present study. Second, the animals were sacrificed immediately after the Y-maze test, one day after general anesthesia. If the observation and sacrifice times in the present study were different, the results might have been different. However, it should be considered that the effect of administered urinary trypsin inhibitor had disappeared within 72 h [37,38]. Third, the age of participants should also be considered since older adults are vulnerable to stress. Fourth, whether ulinastatin can penetrate the BBB was not considered. If ulinastatin can cross the BBB, it may have a direct effect on microglial activity. However, if ulinastatin cannot cross the BBB, the effect of ulinastatin on microglial activity might be indirect via systemic anti-inflammatory effects. This was not determined in the present study because cytokines were evaluated in the brain, not in the blood. Fifth, the ulinastatin dose was not considered. Although 50,000 U of ulinastatin was used in the present study, as it has been the usual dose for anti-inflammatory effects in previous studies, there has been no study on the definite dose response of ulinastatin.

## 4. Materials and Methods

All experiments were performed in accordance with the National Institutes of Health (NIH) Guidelines for the Care and Use of Laboratory Animals. After obtaining approval from the Institutional Animal Care and Use Committee (IACUC) of Konkuk University (approval number: KU22223-1), all experiments were conducted at the Konkuk University Laboratory Animal Research Center in accordance with the IACUC guidelines.


**Experiments preparation**


Male Sprague–Dawley (SD) rats aged 18 months were purchased from Koatech (Pyeongtaek, Republic of Korea). The rats were housed in cages with free access to water and food. The room was maintained with a standard cycle of light and dark every 12 h light/dark cycle, with lights on at 7:00 and lights off at 19:00, at a room temperature of 25 °C. The rats were acclimated to the experimental conditions for 7 days before the study and fed with a standard diet with free access to water.


**Y-maze test**


After acclimation, the Y-maze test was performed to evaluate cognitive function on the day before and one day after general anesthesia. The Y-maze consisted of three arms: A, B, and C. Each arm had a length of 50 cm, height of 25 cm, and width of 10 cm and was connected to the other arms at an angle of 120°. Before the start of the test, all arms were wiped with 70% alcohol, and arm C was blocked with by placing a board at the center of the Y-maze to close the route into arm C. The rats were released from a cage at the end of arm A and allowed to move freely between arms A and B for 15 min, to adapt to the device. After adaptation, the rats were returned to the cage and allowed to rest for 1 h. Next, all arms were wiped with 70% alcohol. The board used to close the route into arm C was removed for free access of the rats to all arms, and the rats were released from the cage at the end of arm A. Movement was recorded using a video camera for 5 min. The frequency of total entries into each arm and the spontaneous alternations were recorded. The alternation ratio was calculated using the formula of [(number of spontaneous alternations)/(total arm entries − 2)] × 100%. If the rat had an alteration ratio <40% on the day before general anesthesia, it was considered to have a preexisting cognitive disorder and was excluded from the study.


**Grouping and general anesthesia**


Rats were randomly allocated into the ulinastatin and control groups. Ulinastatin (50,000 U/mL; Ulistin^®^, Hanlim Pharm., Seoul, Republic of Korea) and normal saline (1 mL) were intraperitoneally administered to the ulinastatin group and control groups, respectively. Anesthesia was induced by intraperitoneal administration of ketamine (100 mg/kg; Yuhan, Seoul, Republic of Korea) and xylazine (10 mg/kg; Sigma-Aldrich, St. Louis, MO, USA). A heating pad was placed on the surgical platform to maintain the body temperature at approximately 37 °C during anesthesia. The rats were secured in the supine position and fastened to the surgical platform for endotracheal intubation, using a 16 G catheter with a length of 45 mm (Dukwoo Medical, Hwaseong, Republic of Korea). The correct position of the catheter for intubation was confirmed by symmetrical chest expansion. The 16 G catheter for endotracheal intubation was connected to a ventilator (Harvard Apparatus, Holliston, MA, USA). The ventilator settings used during anesthesia are as follows: (1) fraction of inspired oxygen, 0.5; (2) inspired flow rate, 150 mL/min; (3) tidal volume, 6 mL/kg; (4) respiration rate, 50 breaths/min; (5) inspiration and expiration ratio, 1:1; and (6) positive end-expiratory pressure, 5 cmH_2_O. Anesthesia was maintained for 2 h using isoflurane (1.5 volume%), after which the vaporizer was switched off. Mechanical ventilation was maintained until full recovery of spontaneous ventilation was confirmed. Next, the 16 G catheter for endotracheal intubation was removed, and the rats were returned to their cages.


**Brain tissue preparation**


After the Y-maze test on one day after general anesthesia, rats were anesthetized using isoflurane (5 volume%) with oxygen (0.3 L/min) and nitrous oxide (0.7 L/min) in the anesthesia induction chamber. The rats were sacrificed after confirmation of successful anesthesia and dissected to expose the abdominal aorta. Administration of 1X phosphate-buffered saline (PBS) into the abdominal aorta was performed until whole blood was exsanguinated and the color of the liver was changed from red to white. To obtain the brain, the skull was opened using scissors and forceps. The brain was divided into two hemispheres. One hemisphere (right side) was used to confirm the degree of inflammation. This hemisphere was transferred into a 2 mL Eppendorf tube^®^ (Eppendorf, Hamburg, Germany) and stored in a deep freezer. The other hemisphere (left side) was used to confirm neuronal damage and microglial activation with phenotypic differentiation. This hemisphere was stored in a 50 mL conical tube, containing 4% paraformaldehyde (PFA; BIOSESANG, Yongin, Republic of Korea).

Neuronal damage was evaluated, using immunofluorescence staining for caspase-1, an indicator of apoptosis. The degree of inflammation was assessed using enzyme-linked immunosorbent assay (ELISA) to detect cytokines. The activation of microglia with phenotypic differentiation was evaluated by immunofluorescence staining. All assessments were performed in the area of hippocampus.


**Neuronal damage**


The left brain that was stored in a conical tube was cut into three pieces and transferred into paraffin cassettes. The paraffin cassettes, containing the tissues, were washed with water for 1 h and rinsed with 4% PFA. The washed paraffin cassettes were transferred to a tissue processor (Leica Biosystems, Wetzlar, Germany) to process the tissues for paraffin sectioning. The tissue processor settings were as follows: formalin1 for 1 h, formalin2 for 1 h, 70% alcohol for 1 h, 80% alcohol for 1 h, 90% alcohol for 1 h, 100% alcohol1 for 1 h, 100% alcohol2 for 1 h, xylene1 for 1 h, xylene2 for 1 h, paraffin1 for 2 h, and paraffin2 for 2 h. After tissue processing, the paraffin cassettes were transferred to tissue embedding centers (Leica Biosystems, Germany) for paraffin embedding. Next, the paraffin cassettes were transferred into a freezer for 30 min to fix the tissues. The tissues were then cut into sections, using a microtome (Leica Biosystems, Germany), and immersed in a warm water chamber at 37 °C. After immersion, the tissues were transferred to glass microscope slides. The tissue on the glass slide was dried on a hotplate at 40 °C for 1 h and 56 °C for 30 min. It was then immersed in xylene1 for 15 min and xylene2 for 10 min to melt the paraffin. Next, it was rehydrated with 100% ethanol1 for 3 min, 100% ethanol2 for 3 min, 90% ethanol for 3 min, 80% ethanol for 3 min, and 70% ethanol for 3 min. The rehydrated tissue was immersed in antigen retrieval and 1X citrate buffer, and then microwaved three times for 5 min. The microwaved tissue on the glass slide was cooled to room temperature for 20 min and treated with 5% goat serum, a blocking solution. Next, the tissue was washed three times with 1X PBS for 5 min. Caspase-1 antibody (Invitrogen, Waltham, MA, USA) was used as the primary antibody to detect neuronal damage. The antibody was diluted to 1/200, using 5% goat serum. The tissue on the microscope glass slide was treated with primary antibody and allowed to react at room temperature for 1 h. The treated tissue was washed three times with 1X PBS for 5 min. A 488 anti-rabbit antibody (Invitrogen, USA) was used as the secondary antibody for target cell staining. The 488 anti-rabbit antibody was diluted to 1/1000 using 5% goat serum. The washed tissue on the glass slide was treated with the secondary antibody and allowed to react in the dark at room temperature for 1 h. The treated tissue was washed three times with 1X PBS for 5 min. For all cells stains, 4′,6-diamidino-2-phenylindole (DAPI; Invitrogen, USA) was used. DAPI was diluted to 1/2000 in distilled water. The washed tissue on the glass slide was treated with DAPI and allowed to react in the dark at room temperature for 5 min. Next, the treated tissue was washed three times with 1X PBS for 5 min. It was then dehydrated with 70% ethanol for 3 min, 80% ethanol for 3 min, 90% ethanol for 3 min, 100% ethanol1 for 3 min, 100% ethanol2 for 3 min, xylene1 for 3 min, and xylene2 for 3 min. After completing the cell staining process, the tissue was mounted using a cover slip with a drop of anti-fade mounting medium (Vector, Newark, NJ, USA). The cover slip was sealed with nail polish to prevent drying and movement under the microscope. The stained tissue on the microscope glass slide was observed using a fluorescence microscope (Olympus, Tokyo, Japan).


**ELISA for detection of cytokines**


The pro-inflammatory cytokines, TNF-α (Abcam, Waltham, MA, USA) and IL-1β (Abcam, USA), and anti-inflammatory cytokines, IL-4 and IL-10, were measured. The right hemisphere of the brain stored in the deep freezer was homogenized and centrifuged at 12,000× *g* for 15 min at 4 °C. After centrifugation, the supernatant was used to detect the cytokines, using an ELISA kit. The cytokine levels were measured using a microplate reader.


**Immunofluorescence staining to detect the activated microglia with microglia with differentiation of the phenotypic expression**


The stored left hemisphere stored in a conical tube was cut into three pieces and transferred into paraffin cassettes. The staining process was the same as that for the evaluation of the neuronal damage.

Ionized calcium-binding adapter molecule 1 (Iba1) antibody (Invitrogen, USA) and transmembrane protein 119 (TMEM 119) antibody (Invitrogen, USA) for microglia; cluster of differentiation (CD)16 antibody (Invitrogen, USA) for M1 phenotype, a pro-inflammatory producer; and CD206 antibody (Invitrogen, USA) for M2 phenotype, an anti-inflammatory producer, were the primary antibodies used for target cell staining. Iba1 was diluted to 1/1000, using 5% goat serum, while other primary antibodies were diluted to 1/200 using 5% goat serum. The 488 anti-rabbit antibody and 594 anti-mouse antibody (Invitrogen, USA) were used as the secondary antibodies for target cell staining. The 488 anti-rabbit antibody was diluted to 1/1000 using 5% goat serum, while the 594 anti-mouse antibody was diluted to 1/2000 using 5% goat serum.


**Statistical analysis**


The primary outcome was the alternation ratio on one day after general anesthesia, while the secondary outcome was microglial activation. From a pilot study with 3 rats in each group, the alternation ratios of 21 ± 25% in the control group and 62 ± 13% in the ulinastatin group, and microglial activation of 42.65 ± 4.68% in the control group and 4.62 ± 0.67% in the ulinastatin group, were checked. The calculated sample sizes of 12 for the primary outcome and 4 for the secondary outcome were determined from the pilot study with an α of 0.05 and power of 0.9.

Statistical analyses were performed using GraphPad Prism software 8.0.1.244 (GraphPad Software, La Jolla, CA, USA). Inter-group statistical significance between the two groups was confirmed by the *t*-test, while intra-group statistical significance was confirmed using a paired *t*-test. All data are expressed as the number of experiments, mean ± standard deviation, or median (interquartile range). A *p*-value less than 0.05 was considered to indicate statistical significance.

## 5. Conclusions

Administration of ulinastatin before general anesthesia prevents neuronal damage and cognitive decline after general anesthesia, in terms of the aspect of behavior, as evaluated by Y-maze test. The protective effect of ulinastatin is associated with the inhibition of microglial activation. Its target among activated microglia is the M1 phenotype.

## Figures and Tables

**Figure 1 ijms-25-02708-f001:**
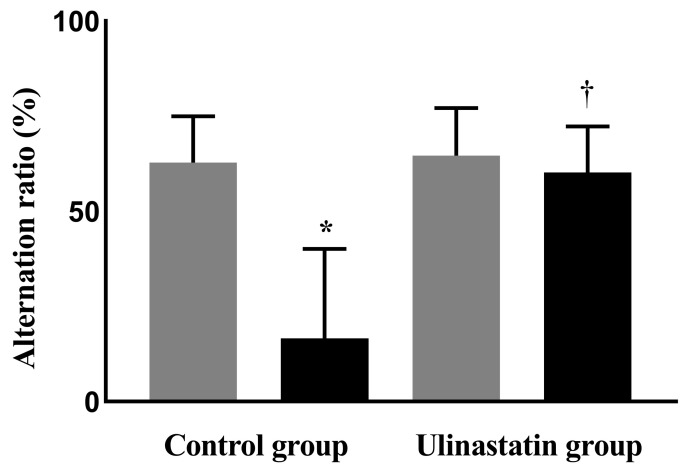
Cognitive function on the day before general anesthesia (gray bar) and one day after general anesthesia (black bar), using the Y-maze test. *: *p* = 0.005, compared with on the day before general anesthesia (gray bar) and one day after general anesthesia (black bar). ^†^: *p* = 0.006, compared with the control group.

**Figure 2 ijms-25-02708-f002:**
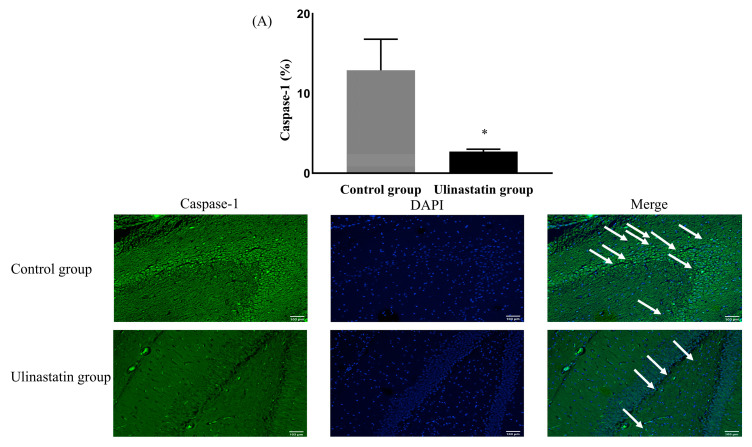
Neuronal damage, using the expression of caspase-1 (represented with white arrows) (**A**) and the number of cells, expressing caspase-1 (represented with white arrows) (**B**). *: *p* = 0.004, compared with the control group ^†^: *p* < 0.001, compared with the control group Abbreviations: DAPI, 4′,6-diamidino-2-phenylindole.

**Figure 3 ijms-25-02708-f003:**
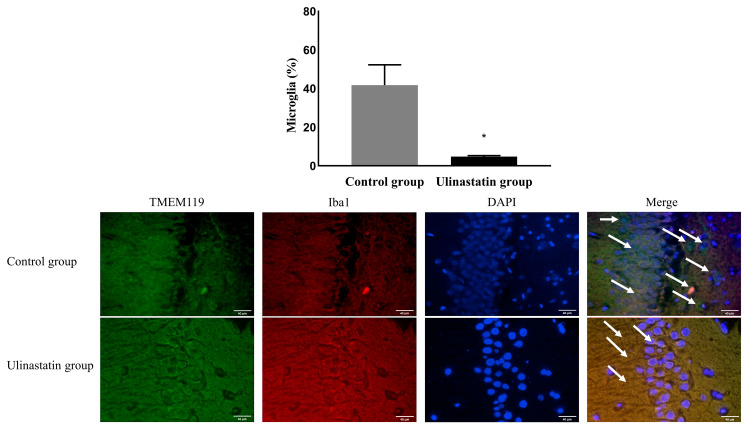
Immunofluorescence staining to detect the activated microglia (represented with white arrows). Abbreviations: TMEM 119, transmembrane protein 119; Iba1, ionized calcium-binding adapter molecule 1; DAPI, 4′,6-diamidino-2-phenylindole. *: *p* < 0.001, compared with the control group.

**Figure 4 ijms-25-02708-f004:**
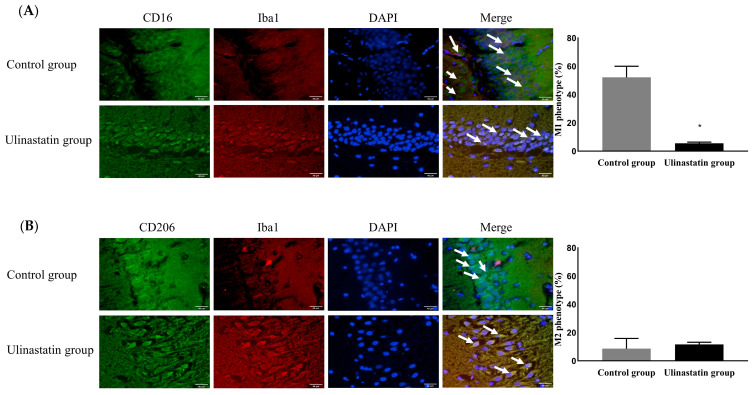
Differentiation of the phenotypic expression of microglia, M1 phenotype (represented with white arrows) as a pro-inflammatory producer (**A**) and M2 phenotype (represented with white arrows) as an anti-inflammatory producer (**B**). Abbreviations: CD, cluster of differentiation; Iba1, ionized calcium-binding adapter molecule; DAPI, 4′,6-diamidino-2-phenylindole. *: *p* < 0.001, compared with the control group.

**Table 1 ijms-25-02708-t001:** Enzyme-linked immunosorbent assay (ELISA) for the detection of cytokines between the control and ulinastatin groups.

	Control Group	Ulinastatin Group	*p* Value
Pro-inflammatory cytokines			
TNF-α (pg/mL)	1456.00 ± 303.60	7.35 ± 0.89	<0.001
IL-1β (pg/mL)	1298.00 ± 783.50	8.54 ± 1.52	0.0062
Anti-inflammatory cytokines			
IL-4 (pg/mL)	23.52 ± 4.06	25.72 ± 0.98	0.28
IL-10 (pg/mL)	7.02 ± 1.80	14.55 ± 1.89	<0.001

Data are expressed as number of experiments, mean ± standard deviation or median (interquartile range). Abbreviations: TNF-α, tumor necrosis factor-α; IL, interleukin.

## Data Availability

The datasets are available from the corresponding author upon reasonable request.

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
