# Peer review of "The Preventive Effect of Urinary Trypsin Inhibitor on Postoperative Cognitive Dysfunction, on the Aspect of Behavior, Evaluated by Y-Maze Test, via Modulation of Microglial Activity"

_ijms, 2024, doi:10.3390/ijms25052708_

Round 1

Reviewer 1 Report

Comments and Suggestions for Authors

1. The introduction does not cover the background of the study properly.

2. Using Y-maze only as a behavioral test is not enough to conclude the cognition-enhancing ability of the test substance. 

3. For the "neuronal damage", the scoring methods used for counting cells must be included. Further, the specific region of the brain used for the histology must be highlighted. Hippocampus and prefrontal cortex are mostly used for behavioral assessments.

4. There are a lot of unwanted information under the subheading "statistics". What is needed others is the description of your statistics e.g. t-test, ANOVA, p values, post hoc etc and the software used.

5. Results are okay, but figure 2 is not clear enough to understand.

6. I am skeptical about the conclusion on cognition as 1 behavioral test is not enough for this big claim by authors.

Comments on the Quality of English Language

English revision required.

Reviewer 2 Report

Comments and Suggestions for Authors

Firstly, I would like to thank the authors for addressing strategies for preventing post-general anesthesia cognitive complications. This is an area of interest in the current context of increased access to invasive techniques that necessitate general anesthesia, particularly in advancing age groups.

However, I do have some doubts/aspects to consider for the authors before evaluating its potential for publication:

  1. ABSTRACT: Clearly state from the outset in the first sentence that it is an experimental study with 1:1 randomization; in the methodology section, provide an explanation of the number of animals per group.
  2. INTRODUCCIÓN: The first sentence does not seem entirely accurate. It is necessary to provide a better explanation of the relationship between the increased need for surgery or other invasive techniques requiring general anesthesia and advancing age. Additionally, it is essential to delve deeper into the risk factors for the development of POCD, considering not only prior cognitive impairment and its severity but also factors such as polypharmacy, substance consumption, and details about the surgery itself, including its type and duration, if there are complications during or after surgery, the hemodynamic stability or instability during the process should also be addressed. How does POCD differ from an acute confusional syndrome induced by an infection or other organic cause?

    On the other hand, doesn't it seem a bit simplistic to directly associate POCD with neuroinflammation and overlook other mechanisms, such as hemodynamic factors, including the undisputed role of hypoxia?

    3. METHODOLOGY: Explain the two-phase design: a pilot phase followed by the discovery phase. Why choose such young animals? Is the aim to extrapolate the results to a population aged 45 or older, or is the target population 65 or older? I mention this because in a healthy subject aged 45 without surgery-related incidents, the risk of clinically relevant POCD is relatively low. Have you considered conducting different types of surgeries with varying anesthesia protocols - which differ based on the type of anesthesia - and based on hemodynamic stability or instability during the process? These are aspects that can be induced in animal models and add value to such studies. Additionally, why opt for intraperitoneal administration? (Is there no safer and more translatable route for future human application?)

    Why not wait longer before conducting the neuropathological evaluation? It would be interesting to assess the influence on clinical outcomes and measurable biomarkers in life regarding the administration or non-administration of the drug. Were commonly used inflammatory biomarkers in humans, such as C-reactive protein, erythrocyte sedimentation rate, measured in the blood? Were neuroinflammation biomarkers with good performance in plasma, like GFAP, or neurodegeneration biomarkers with good performance in plasma, such as NfL, measured? For example, it is well-known that individuals with higher plasma NfL levels in the acute phase of acute confusional syndrome have a worse vital and functional prognosis (data already available in humans).

    4.

    DISCUSSION AND CONCLUSIONS: I believe there is not enough evidence provided to assert that the benefit on neuroinflammation is demonstrated. I consider that such definitive statements diminish the value of the work. From my perspective, the study explores how anesthesia may impact neuroinflammation mechanisms and simultaneously explores the potential to modulate these changes. It is a highly exploratory study, far from being representative of what would be done in future clinical practice. Additionally, it solely investigates neuroinflammation and neglects other equally relevant mechanisms, without allowing time to assess potential adverse effects after medication administration that could be potentially serious.

    I think the discussion should review not only similar experiences related to the explored condition but also acute confusional syndromes of other etiologies, highlighting differences between findings in animals and potentially in humans in the future. In other words, there should be a more critical examination of the results, which does not diminish their value but rather enhances it. Finally, I would add more information about ideas for future studies in the same line of research.

Round 2

Reviewer 1 Report

Comments and Suggestions for Authors

One behavioural test is not enough for a research title like this "The preventive effect of urinary trypsin inhibitor on postopera- 2 tive cognitive dysfunction via modulation of microglial activity".

If you insist, then change your title to something else.

Author Response

At first, I thank the editors and referees of the “International Journal of Molecular Sciences” for taking their times to review of my paper, entitled “The preventive effect of urinary trypsin inhibitor on postoperative cognitive dysfunction via modulation of microglial activity”.

I have made some corrections and clarifications in the manuscript after going over the referee’s comments. The changes are summarized below and the corrected or newly added sentences were expressed with red-color in the manuscript.

Reviewer-1

  1. One behavioural test is not enough for a research title like this "The preventive effect of urinary trypsin inhibitor on postopera- 2 tive cognitive dysfunction via modulation of microglial activity". If you insist, then change your title to something else.

: We absolutely agreed with Reviewer-1 opinion that just one behavioral test, Y-maze test, was not enough to evaluate cognitive function. However, as we described in Discussion, previous reports have demonstrated that Y-maze test is a useful tool to evaluated cognitive function. As Reviewer-1’s opinion, we changed the title, “The preventive effect of urinary trypsin inhibitor on postoperative cognitive dysfunction, on the aspect of behavior, evaluated by Y-maze test, via modulation of microglial activity”. We also changed the sentence in Background of Abstract.

Reviewer-2

  1. I appreciate the authors for incorporating the improvement suggestions provided. I believe this has significantly enhanced both the introduction and discussion sections. The remaining limitations mentioned, which are more methodological in nature (such as the age of the animals, lack of determination of specific neuroinflammation biomarkers...), could not be addressed as they were not considered at the study design stage. However, the authors have improved the wording to indicate that the results are exploratory and not clearly generalizable, to be corroborated in other studies where methodological changes as suggested would be considered. Therefore, at this time, I have no further suggestions/comments to present

: We thank for informative comments from Reviewer-2.

I hope the revised manuscript will better meet the requirements of the “International Journal of Molecular Sciences” for publication.

I thank you again for the constructive review by the referees.

Sincerely yours,

Seong-Hyop Kim, M.D., Ph.D.

Reviewer 2 Report

Comments and Suggestions for Authors

I appreciate the authors for incorporating the improvement suggestions provided. I believe this has significantly enhanced both the introduction and discussion sections. The remaining limitations mentioned, which are more methodological in nature (such as the age of the animals, lack of determination of specific neuroinflammation biomarkers...), could not be addressed as they were not considered at the study design stage. However, the authors have improved the wording to indicate that the results are exploratory and not clearly generalizable, to be corroborated in other studies where methodological changes as suggested would be considered. Therefore, at this time, I have no further suggestions/comments to present

Author Response

(The authors gave the same response as above.)

Round 3

Reviewer 1 Report

Comments and Suggestions for Authors

Change the conclusion to match the title of the manuscript. 

Author Response

At first, I thank the editors and referees of the “International Journal of Molecular Sciences” for taking their times to review of my paper, entitled “The preventive effect of urinary trypsin inhibitor on postoperative cognitive dysfunction via modulation of microglial activity”.

I have made some corrections and clarifications in the manuscript after going over the referee’s comments. The changes are summarized below and the corrected or newly added sentences were expressed with red-color in the manuscript.

Reviewer-1

  1. Change the conclusion to match the title of the manuscript.

: We changed the conclusion, as Reviewer’s comment.

→ Administration of ulinastatin before general anesthesia prevents neuronal damage and cognitive decline after general anesthesia, on the aspect of behavior, evaluated by Y-maze test. The protective effect of ulinastatin is associated with the inhibition of microglial activation. Its target among activated microglia is the M1 phenotype.

I hope the revised manuscript will better meet the requirements of the “International Journal of Molecular Sciences” for publication.

I thank you again for the constructive review by the referees.

Sincerely yours,

Seong-Hyop Kim, M.D., Ph.D.